# Pre-Exercise Blood Glucose Levels Determine the Amount of Orally Administered Carbohydrates during Physical Exercise in Individuals with Type 1 Diabetes—A Randomized Cross-Over Trial

**DOI:** 10.3390/nu11061287

**Published:** 2019-06-06

**Authors:** Othmar Moser, Max L. Eckstein, Alexander Mueller, Philipp Birnbaumer, Felix Aberer, Gerd Koehler, Caren Sourij, Harald Kojzar, Peter Pferschy, Pavel Dietz, Richard M. Bracken, Peter Hofmann, Harald Sourij

**Affiliations:** 1Division of Endocrinology and Diabetology, Department of Internal Medicine, Medical University of Graz, 8036 Graz, Austria; max.eckstein@swansea.ac.uk (M.L.E.); felix.aberer@medunigraz.at (F.A.); gerd.koehler@klinikum-graz.at (G.K.); caren.sourij@medunigraz.at (C.S.); harald.kojzar@medunigraz.at (H.K.); peter.pferschy@medunigraz.at (P.P.); ha.sourij@medunigraz.at (H.S.); 2Applied Sport, Technology, Exercise and Medicine Research Centre (A-STEM), College of Engineering, Swansea University, Swansea SA18EN, UK; r.m.bracken@swansea.ac.uk; 3Exercise Physiology, Training and Training Therapy Research Group, Institute of Sports Science, University of Graz, 8010 Graz, Austria; alexander.mueller@uni-graz.at (A.M.); philipp.birnbaumer@uni-graz.at (P.B.); peter.hofmann@uni-graz.at (P.H.); 4Institute of Occupational, Social and Environmental Medicine, University Medical Centre of the University of Mainz, 55161 Mainz, Germany; pdietz@uni-mainz.de; 5Diabetes Research Group, Medical School, Swansea University, Swansea SA28AP, UK

**Keywords:** carbohydrates, exercise, type 1 diabetes, euglycemia, insulin

## Abstract

The aim of the study was to assess the amount of orally administered carbohydrates needed to maintain euglycemia during moderate-intensity exercise in individuals with type 1 diabetes. Nine participants with type 1 diabetes (four women, age 32.1 ± 9.0 years, BMI 25.5 ± 3.9 kg/m^2^, HbA_1c_ 55 ± 7 mmol/mol (7.2 ± 0.6%)) on insulin Degludec were randomized to cycle for 55 min at moderate intensity (63 ± 7% VO_2peak_) for five consecutive days on either 75% or 100% of their regular basal insulin dose. The impact of pre-exercise blood glucose concentration on the carbohydrate requirement was analyzed by one-way ANOVA stratified for pre-exercise blood glucose quartiles. The effect of the basal insulin dose on the amount of orally administered carbohydrates was evaluated by Wilcoxon matched-pairs signed-rank test. The amount of orally administered carbohydrates during the continuous exercise sessions was similar for both trial arms (75% or 100% basal insulin) with median [IQR] of 36 g (9–62 g) and 36 g (9–66 g) (*p* = 0.78). The amount of orally administered carbohydrates was determined by pre-exercise blood glucose concentration for both trial arms (*p* = 0.03). Our study elucidated the importance of pre-exercise glucose concentration related orally administered carbohydrates to maintain euglycemia during exercise in individuals with type 1 diabetes.

## 1. Introduction

The beneficial effects of physical activity and exercise are well described in individuals with type 1 diabetes [1,2]. Regular physical activity and exercise improve cardiovascular risk factors, insulin sensitivity, body mass index (BMI), waist circumference as well as quality of life [3]. Even in individuals with type 1 diabetes suffering from chronic kidney disease, leisure time physical activity is associated with a lower risk of mortality [4]. Supporting this evidence, current guidelines recommend 150 min per week of moderate-intensity physical exercise or 75 min per week of vigorous intensity physical exercise [1]. However, individuals with type 1 diabetes may resist performing physical exercise due to the risk of glycemic disturbances that involve both hypo- and hyperglycemia [5]. In previous studies, the emphasis has been on how to avoid exercise-induced hypoglycemia by means of pre-exercise bolus insulin dose reductions for both continuous [6] and high-intensity interval exercises [7]. Furthermore, insulin Glargine U-100 (Sanofi, FRA) and insulin Detemir (Novo Nordisk A/S, DEN) basal dose reductions were also found to prevent hypoglycemia over 24 h after performing an acute bout of physical exercise [8]. A less commonly applied, but highly effective approach to reduce the risk of exercise-induced hypoglycemia faces subcutaneous mini-dose injections of glucagon [9]; in this study Rickels and colleagues showed that 150 mg of glucagon may be more effective than a 50% basal insulin reduction for preventing exercise-induced hypoglycemia and may result in less post-intervention hyperglycemia than a standardized ingestion of 40 g oral glucose tablets. The only known exogenous approach to avoid post-exercise hypoglycemia is by means of a 10-sec maximal sprint after a moderate-intensity exercise session [10]. Via this short bout of high-intensity exercise, counterregulatory hormones adrenaline and noradrenaline increase hepatic glucose production, hence leading to a stabilization in blood glucose concentration. Bolus insulin dose reductions are often linked to pre-exercise hyperglycemia [11,12] that might impact on improvements in glycemic control (HbA_1c_) and potentially increase the risk of ketoacidosis related to insulin omissions [13]. If individuals with type 1 diabetes have poor glycemic control oxygen economy as well as heart rate dynamics are impaired [14,15]. Consequently, good diabetes management includes both, physical exercise and good glycemic control to induce a beneficial effect for the individual with type 1 diabetes.

These studies and a recent consensus statement suggest that individuals with type 1 diabetes should aim to exercise in a euglycemic range (e.g., 7.0–10.0 mmol/L [126 mg/dL–180 mg/dL]) to evoke positive exercise-induced treatment effects [8]. 

Blood glucose decrease during physical exercise is depending on various endogenous and exogenous factors like fitness level, c-peptide status, hormonal responses, glucose transporter type 4 efficacy as well as type of therapy, insulin on board, external workload, type of exercise etc. Initially, research showed a linear blood glucose decrease during endurance exercise depending on the mean exercise intensity [11]. Intriguingly, it was recently shown that the decline in blood glucose levels during physical exercise is associated with pre-exercise blood glucose levels in children with type 1 diabetes (‘the higher the start, the harder they fall’) [16]. The undesired side-effect of physical exercise was also illustrated within this study: if the drop in the blood glucose concentration was not treated with orally administered carbohydrates, the incidence of exercise-induced hypoglycemia was ~44%. 

Current guidelines outline how to adapt bolus and basal insulin therapy depending on different volumes and types of exercise [1,8]. However, there is a dearth of information on the required amount of orally administered carbohydrates to maintain euglycemia during physical exercise in individuals with type 1 diabetes. Therefore, the aim of the present study was to investigate the amount of orally administered carbohydrates needed to maintain euglycemia during moderate-intensity physical exercise, taking pre-exercise blood glucose levels and the basal insulin dose into account. Additionally, we aimed to assess changes in carbohydrate demand over consecutive days of exercise testing (exercise accumulating effect). 

## 2. Materials and Methods

### 2.1. Participants

Potential participants were recruited via telephone; contact details were provided from a volunteers-database from the Medical University of Graz, Austria for individuals with type 1 diabetes. Inclusion criteria were defined as: individuals with type 1 diabetes (18–65 years, both including, male and female, diagnosed >12 months) on multiple daily injections with insulin Degludec (Novo Nordisk A/S, DEN) as a basal insulin, body mass index between 18.5 and 32 kg/m^2^, glycated hemoglobin (HbA_1c_) from 42 to 86 mmol/mol (6 to 10%) and fasting c-peptide level of <0.25 nmol/L. Furthermore, participants had to be regularly physically active according to the recommendations of the American College of Sports Medicine (ACSM) [17]. These guidelines define regular physical activity between 150 min of moderate physical activity per week (~65% maximum oxygen uptake [VO_2max_]) or 75 min of vigorous physical activity (~80% VO_2max_). Once potential participants were screened for eligibility, physical activity was assessed via the short International Physical Activity Questionnaire (IPAQ-SF). Following this procedure, randomization was applied via the “Randomizer for Clinical Trials”, which was provided from the Institute of Medical Informatics, Statistics and Documentation, Medical University of Graz [18].

This randomized cross-over trial study was approved by the ethics committee of the Medical University of Graz, Austria (registry number: 26-334 ex 16/17) and local health authorities. Participants gave their written informed consent before any trial related activities. All procedures in this trial were performed according to Good Clinical Practice (GCP) and the Declaration of Helsinki (DoH). Individuals that would have had the potential to interfere or bias the study results were not included as described in the clinical registration (DRKS.de: DRKS00013477; EudraCT: 2017-000922-37) and the primary outcome was published recently [19]. Within this paper, we present secondary data analysis of the study. Overall, the study consisted of 14 visits for each participant whereas visit 1, 7 and 14 were not exercise related. At visit 7 and 14, participants were asked to come to the lab 24-h post-exercise for a venous blood sample, which was unrelated for the outcomes of this study [19].

### 2.2. Peak Cardio-Pulmonary Exercise (CPX) Testing

At the first visit, participants performed a peak cardio-pulmonary exercise test until volitional exhaustion on a cycle ergometer as outlined previously [7]. 

During cardio-pulmonary exercise testing, heart rate was measured continuously via chest belt telemetry and averaged over 5 s (s810i, Polar Electro, Espoo, Finland). A 12-lead electrocardiogram (ECG) and blood pressure measurement were obtained for cardiovascular monitoring (ZAN 600; nSpire Health, Oberthulba, Germany). Throughout cardio-pulmonary exercise testing, breath-by-breath measurements were conducted and averaged over 5 s for later analysis (METAMAX^®^ 3B; Cortex Biophysik GmbH, Leipzig, Germany). 

Capillary blood lactate and blood glucose samples (20 μL) were taken from the earlobe minimal-invasively prior to the test, during the warm-up, after every increment, during active cool-down and passive cool-down (Biosen S-line, EKF Diagnostics, Barleben, Germany). The aim of the cardio-pulmonary exercise test was to determine the peak power output (P_peak_) and the first (LTP_1_) and second lactate turn points (LTP_2_), that were used to prescribe the exercise intensity for the following exercise periods [20].

Peak cardio-pulmonary exercise testing was terminated at the point of maximum exhaustion that was defined as a peak lactate concentration of more than 10 mmol/L, a respiratory exchange ratio above 1.1, a plateau in the oxygen uptake (VO_2_) or the inability to maintain a pedaling cadence of more than 50 rpm for 5 s. 

During all cardio-pulmonary exercise tests, additional blood glucose samples were taken from the fingertip (6 μL) at a 2-min interval to reduce the risk of a potential hypoglycemic event (FreeStyle Libre, Abbott, UK).

### 2.3. Diabetes Specificities

Participants included in this study were on a stable insulin Degludec therapy (Novo Nordisk A/S, DEN) for least 3 months prior to their study inclusion. A stable basal insulin therapy was defined as a pre-breakfast self-measured blood glucose concentration of 4 to 7 mmol/L (72–126 mg/dL) over six consecutive days with the same basal insulin dose. Three days prior to the beginning of the exercise procedure participants were informed to either maintain (100%) or reduce (75%) their regular basal insulin therapy following the randomization procedure (Figure 1). Participants injected their basal insulin every day in the evening. Three participants were using insulin lispro (Eli Lilly, Indianapolis, IN, USA) while six participants were using insulin aspart (Novo Nordisk A/S, Bagsværd, Denmark) as a bolus insulin. 

During the course of the trial (Visit 1 to Visit 14), participants were provided with an unblinded intermittently viewed continuous glucose monitoring (iCGM) system (FreeStyle Libre, UK) to facilitate glucose monitoring. Additionally, participants were given spare sensors that if in the case sensors were lost, expired or lost function, participants could replace them themselves. Participants were trained by the health care professional on how to use the device and how to overcome potential system inaccuracies within the first 24-h calibration period to reduce the risk of diabetes or exercise-related adverse events during the run of the trial [19].

Glycemic ranges were reported according to the recent position statement by the American Diabetes Association [21]. Hypoglycemia was divided in two subsections: a glucose alert value level 1, ≤3.9 mmol/L (≤70 mg/dL) and clinically important hypoglycemia level 2, <3.0 mmol/L (<54 mg/dL). Euglycemia was defined as the range between 4.0 (72 mg/dL) and 9.9 mmol/L (179 mg/dL) while hyperglycemia was defined as >10 mmol/L (>180 mg/dL).

### 2.4. Moderate-Intensity Physical Exercise Periods

Prior to the first day of the exercise periods, participants were asked to consume between 5 to 7 g carbohydrates per kilogram of body weight per day and to replicate this diet for both dosing schemes [22]. Participants were informed to inject the last bolus insulin and consume the last pre-exercise carbohydrate-rich meal at least 2 h before exercise, to reduce the impact of glycemic variability during the course of the exercise sessions.

Following the analysis of the cardio-pulmonary exercise testing and randomization to either a 75% or 100% insulin Degludec dose, participants performed daily exercise sessions at a moderate intensity, defined as the midpoint between the first (LTP_1_) and the second lactate turn point (LTP_2_) (63 ± 7% VO_2peak_), for 5 consecutive days at the clinical research facility of the Medical University of Graz, Austria. The exercise sessions were conducted in the evening at either 5.00 pm or 6.30 pm and lasted for 55 min on an electronically braked cycle ergometer. Participants were allowed to start the exercise session if blood glucose concentration was above 7 mmol/L (126 mg/dL). The start of the exercise session was delayed if the blood glucose concentration was below 7 mmol/L. As a consequence, 15–30 g of oral carbohydrates were given either via glucose gel (67% glucose and 33% fructose) or fruit juice (72% glucose and 28% fructose), which was repeated after 10 min if blood glucose did not increase accordingly following the first anti-hypoglycemic treatment. The amount of orally administered carbohydrates was documented in the case report form (CRF). However, if participants had a pre-exercise blood glucose concentration >15 mmol/L (270 mg/dL) ketone level was measured (FreeStyle Libre, Abbott, UK). If the ketone level was below 1.5 mmol/L participants were allowed to start the exercise session.

During the exercise sessions carbohydrates were administered orally (15–30 g) if the participant dropped below a pre-defined threshold of 7.0 mmol/L (126 mg/dL). If the blood glucose concentration did not increase above 7.0 mmol/L (126 mg/dL) within 10 min after this treatment, this procedure was repeated (every 10 min) as often as required. However, exercise was discontinued if the participant dropped below a threshold of 4.0 mmol/L (70 mg/dL) and was only continued when blood glucose increased via orally administered carbohydrates above 7 mmol/L (126 mg/dL). During every exercise session, cardio-pulmonary parameters were collected similarly to the measurement procedures during the peak cardio-pulmonary exercise test. However, blood lactate and blood glucose parameters were collected by minimal-invasively means after the pre-exercise rest period, the warm-up, every seven minutes during the target workload, after the active cool-down and after the passive cool-down to monitor glycemia. After a minimum wash-out period of 4 weeks, participants performed the second exercise period with the remaining basal insulin dosing scheme.

### 2.5. Statistical Analyses

Sample size estimation was based on the primary outcome from this clinical trial and was defined as time spent in euglycemia over 5 days, assuming that the mean time in euglycaemia is increased by 10% in the 75% insulin Degludec dosing scheme [19].

Data were tested for distribution via Kolmogorov-Smirnoff test. The effect of the pre-exercise blood glucose concentration on orally administered carbohydrates, when stratified for pre-exercise blood glucose quartiles based on median, 25th and 75th percentile, was analyzed via Kruskal–Wallis ANOVA. The effect of the insulin Degludec dose on pre-exercise and post-exercise blood glucose values was calculated via student’s *t*-test or Wilcoxon matched-pairs signed-rank test while the absolute amount of given carbohydrates during exercise sessions in comparison of dosing schemes was calculated via Wilcoxon matched-pairs signed-rank test. The exercise accumulating effect from the first to the last exercise session on carbohydrate consumption during exercise was analyzed via repeated-measures one-way ANOVA.

Data were analyzed via Prism version 7 (GraphPad, San Jose, CA, USA) and significance level was set at *p* ≤ 0.05.

## 3. Results

From the 11 individuals screened for this study, 10 participants were included in the study. Nine participants completed both trial arms since one participant left the study after the first trial arm due to personal reasons. Consequently, the data of this participant was not included in the analysis and results section (per protocol analysis). 

### 3.1. Participant Characteristics and Performance Data

Four women and five men completed the study and had a mean ± standard deviation (SD) age of 32.1 ± 9 years, a body mass index of 25.5 ± 3.8 kg/m^2^, HbA_1c_ of 55 ± 7 (43–62) mmol/mol (7.2 ± 0.6 (6.1–7.8%)), a mean ± SD (min–max.) diabetes duration of 19 ± 11 (6–42) years. All participants achieved the predefined exhaustion criteria of the peak cardio-pulmonary exercise test. Functional capacity obtained from the peak cardio-pulmonary exercise testing is displayed in Table 1. Also, exercise performance values of the continuous exercise test can be seen in Table 2.

### 3.2. Diabetes-Specific Results

During the course of the intervention, total daily dose of insulin was 0.46 ± 0.15 IU/kg body weight while total daily dose of basal insulin was 0.25 ± 0.06 IU/kg body weight for the 100% insulin Degludec arm and 0.19 ± 0.05 for the 75% insulin Degludec arm. Time spent in specified glycemic ranges assessed via capillary blood glucose concentration (Biosen S-line, EKF Diagnostics, Barleben, GER) during continuous exercise sessions based on the basal insulin dose (75% or 100%) is displayed in Figure 2.

The amount of carbohydrates consumed at the last pre-exercise meal was similar between both trial arms (100% insulin Degludec dose 62 ± 6 g versus 75% insulin Degludec dose 64 ± 15 g, *p* = 0.83). The dose of the last pre-exercise bolus insulin dose injection (100% insulin Degludec dose 6 (4–9) IU versus 75% insulin Degludec dose 6 (4–8) IU, *p* = 0.82) as well as the point of time when the bolus insulin was injected until starting the exercise sessions (100% insulin Degludec dose 4 h 36 min ± 53 min versus 75% insulin Degludec dose 4 h 06 min ± 29 min, *p* = 0.22) were similar between both trial arms. Pre-exercise blood glucose concentration for the continuous exercise sessions was 9.86 ± 0.79 mmol/L (177 ± 14 mg/dL) for the 75% insulin Degludec dose and 9.64 ± 0.59 mmol/L (173 ± 10 mg/dL) for the 100% insulin Degludec dose (*p* = 0.66). Post-exercise blood glucose values had a median (interquartile range) of 5.83 mmol/L (5.05–7.57 mmol/L) (105 mg/dL (91–136 mg/dL)) for the 100% insulin Degludec dose and 6.68 mmol/L (5.64–7.19 mmol/L) (120 mg/dL (101–129 mg/dL)) for the 75% insulin Degludec dose (*p* = 0.57). The amount of orally administered carbohydrates to maintain euglycemia during the continuous exercise sessions was also similar for both trial arms with a median (interquartile range) of 36 g (9–62 g) and 36 g (9–66 g) (*p* = 0.78). Furthermore, we did not identify an exercise accumulating effect, since there was no increase in orally administered carbohydrates during the exercise sessions (day 1 until day 5) for both 75% (*p* = 0.83) and 100% (*p* = 0.69) of insulin Degludec.

The effect of pre-exercise blood glucose on the amount of orally administered carbohydrates stratified for pre-exercise blood glucose quartiles is displayed in Table 3.

## 4. Discussion

This is the first study investigating the interaction between pre-exercise blood glucose levels and the amount of orally administered carbohydrates during moderate-intensity exercise, depending on the dose of basal insulin in individuals with type 1 diabetes running on multiple daily injections. A higher pre-exercise blood glucose level resulted in less orally administered carbohydrates to maintain euglycemia. The responsible mechanism behind this occurs on systemic level, at which fast-available energy sources (=blood glucose) are metabolized immediately, and energy source is shifted towards lipolysis in healthy individuals [23]. However, the continuous blood glucose lowering effect induced by exogenous (basal) insulin (=permanent glycolysis) circumvents lipolysis as a main cellular fuel in individuals with type 1 diabetes (Table 2). 

Physical exercise requires individual therapy adaptations in individuals with type 1 diabetes. Individuals that are aiming to lose body weight should reduce their pre- and post-exercise bolus insulin dose depending on the mean exercise intensity [7,11]. In the particular case when physical exercise is performed over consecutive days, the basal insulin dose should also be reduced [19]. However, when aiming to maintain body weight and increase functional capacity (e.g.: improvements in oxygen economy, muscle mass or external workload) additional carbohydrates should be consumed. Furthermore, consuming carbohydrates during physical exercise preserves spontaneous physical exercise like in “healthy individuals” without the need to prepare for upcoming physical activities at the pre-exercise meal. Nonetheless, both strategies face different disadvantages and must be balanced against individual therapies: reducing bolus and/or basal insulin and its association with hyperglycemia might imply a harmful impact on micro- and macrovascular complications [7,11,13]. Consuming additional carbohydrates involves several aspects that need to be considered at the start of exercise, hence requiring profound knowledge about pre-exercise blood glucose concentration, insulin on board, expected blood glucose response depending on the type and volume of physical exercise and others [8,24]. 

In contrast to our findings, Riddell et al. [8] suggest that 10–15 g/h of orally administered carbohydrates protect against hypoglycemia and maintain euglycemia for moderate-intensity physical exercise, however, this assumption was mainly based on experience. Our study suggests that the mean amount of additionally required carbohydrates is higher, however, depending on the blood glucose level at the beginning of the exercise session. Both groups, independent of a full (100%) or a reduced (75%) basal dose needed in median additionally 36 g of carbohydrates per hour during exercise. The wide variability of consumed carbohydrates is closely related to the pre-exercise blood glucose concentration accompanied with low amounts of bolus insulin on board (at least two hours post-bolus insulin injection). 

From a clinical point of view, our results may serve as a first hint on how individuals with type 1 diabetes can maintain euglycemia based on orally administered carbohydrates during ~1 h of moderate-intensity exercise with little bolus insulin on board. It is important to note that carbohydrates were given already at a blood glucose concentration of 7 mmol/L (126 mg/dL) to avoid any clinically relevant hypoglycemia (<3.0 mmol/L; 54 mg/dL) [21]. This early treatment is necessary due to the high rate of change in blood glucose concentration (>0.1 mmol/L; 2 mg/dL/min) during moderate-intensity exercise [7,11,25]. Additionally, it must be taken into account that the glucose threshold of 7 mmol/L (126 mg/dL) is based on the glucose concentration in the blood stream, hence our results cannot be transferred to interstitial glucose concentrations measured via continuous glucose monitoring systems. Based on previous research we assume that orally administered carbohydrates need to be given earlier due to physiological and device-specific lag times between the blood stream and interstitium potentially causing device-specific measurement errors [26,27,28]. 

Our study is limited by the low number of participants, which we tried to overcome by multiple exercise sessions (n = 90). Additionally, even though our participants were trained on carbohydrate counting, a few pre-exercise bolus insulin under-dosages were observed, contributing to a time spent in hyperglycemia of 17.2% of total time (Figure 2). However, such elevated pre-exercise blood glucose levels are often seen in individuals with type 1 diabetes [16] hence indicating a transferability of our study results to the real-life condition of our study.

## 5. Conclusions

Our study elucidated the importance of orally administered carbohydrates to maintain euglycemia during moderate-intensity exercise when pre-exercise bolus insulin was not reduced in individuals with type 1 diabetes. The amount of orally administered carbohydrates is linked to the pre-exercise blood glucose concentration and is not significantly different in comparison between a regular and 25% reduced basal insulin dose and no exercise accumulating effect was found. It is important to note, that no clinically relevant hypoglycemia (<3.0 mmol/L; 54 mg/dL) occurred in 90 exercise sessions with a duration of ~1 h when carbohydrates were administered at a blood glucose concentration of 7 mmol/L (126 mg/dL).

## Figures and Tables

**Figure 1 nutrients-11-01287-f001:**
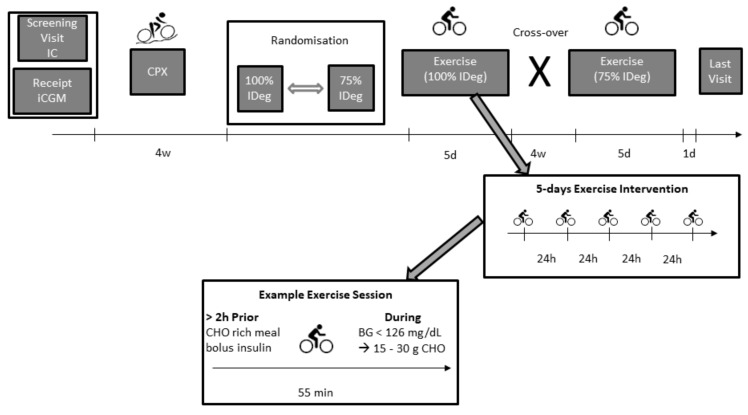
Study flow chart. IC = informed consent procedure, iCGM = intermittently-viewed continuous glucose monitoring system, CPX = peak cardio-pulmonary exercise test, IDeg = insulin Degludec, CHO = carbohydrates, BG = blood glucose.

**Figure 2 nutrients-11-01287-f002:**
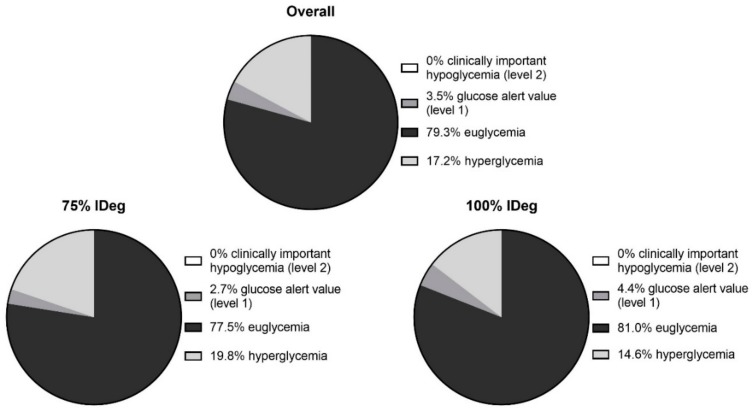
Time spent in glycemic ranges assessed via capillary blood glucose concentration (Biosen S-line, EKF Diagnostics, Barleben, GER) during the continuous exercise sessions for all participants. 75% IDeg: 25% reduced dose of insulin Degludec. 100% IDeg: regular dose of insulin Degludec.

**Table 1 nutrients-11-01287-t001:** Peak cardio-pulmonary exercise test variables.

Variable	Mean ± SD	Min–Max
P_peak_ (Watt/kg)	3.06 ± 0.89	1.57–4.34
P_LTP1_ (Watt/kg)	0.78 ± 0.31	0.25–1.40
P_LTP2_ (Watt/kg)	1.97 ± 0.56	0.98–2.73
VO_2peak_ (mL/kg/min)	39 ± 12	22–64
VO_2LTP1_ (mL/kg/min)	16 ± 6	9–28
VO_2LTP2_ (mL/kg/min)	29 ± 10	14–46
VE_peak_ (L/min)	112 ± 36	75–173
O_2_ pulse_peak_ (100 mL/b/kg)	20.6 ± 5.7	13.9–34.0

P: Power; LTP_1_: Lactate turn point 1; LTP_2_: Lactate turn point 2; VO_2_: Oxygen consumption; VE: Ventilation; O_2_: oxygen; mL: Milliliter; kg: Kilogram; min: Minutes; b: Beats.

**Table 2 nutrients-11-01287-t002:** Continuous exercise testing variables.

Variable	75% IDeg	100% IDeg	*p*-Value
VO_2_ of VO_2peak_ %	64 ± 7	62 ± 6	0.14
HR of HR_peak_ %	74 ± 6	73 ± 7	0.11
VE of VE_peak_ %	43 ± 8	43 ± 8	0.91
RER of RER_peakx_ %	75 ± 4	78 ± 2	0.09
Lactate of Lactate_peak_ %	20 ± 9	19 ± 7	0.57
O_2_ pulse of O_2_pulse_peak_ %	61 ± 8	61 ± 19	0.94

VO_2peak_: Peak oxygen consumption; VE: Ventilation; HR: Heart rate; VE: Ventilation; RER: Respiratory exchange ratio; O_2_: Oxygen.

**Table 3 nutrients-11-01287-t003:** Pre-exercise blood glucose quartiles based on median, 25th and 75th percentile with concomitant carbohydrate consumption.

Quartiles						
Variable		Min	25	75	Max	*p*-Value
75% IDeg	BG (mmol/L) BG (mg/dL)	6.7 ± 0.7 ^†^ 120 ± 13	8.4 ± 0.6 150 ± 11	10.5 ± 0.5 189 ± 9	13.9 ± 2.6 ^†^ 250 ± 46	0.03
	CHO (g)	54 (18–73)	44 (33–72)	36 (18–54)	0 (0–18)	
100% IDeg	BG (mmol/L) BG (mg/dL)	7.3 ± 0.8 * 131 ± 15	8.7 ± 0.3 156 ± 6	10.2 ± 0.5 * 184 ± 10	12.4 ± 1.7 223 ± 30	0.03
	CHO (g)	62 ± 36	40 ± 30	22 ± 25	30 ± 35	

CHO: carbohydrate; IDeg: insulin Degludec; BG: blood glucose; g: Gram. * indicates statistically significant difference (*p* < 0.05) in comparison of Min to 75 within the 100% IDeg arm. ^†^ indicates significance (*p* = 0.05) in comparisons of Min to Max within the 75% IDeg arm.

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
