# Peer review of "Pre-Exercise Blood Glucose Levels Determine the Amount of Orally Administered Carbohydrates during Physical Exercise in Individuals with Type 1 Diabetes—A Randomized Cross-Over Trial"

_nutrients, 2019, doi:10.3390/nu11061287_

Reviewer 1 Report

This manuscript presents the secondary objectives of a published study that primarily explored the effect of basal insulin reduction on glucose control during physical activity in MDI T1D patients. The secondary objectives presented here was to investigate the amount of orally administered carbohydrates needed to maintain euglycemia during moderate-intensity physical exercise, taking pre-exercise blood glucose levels and the basal insulin dose into account. The main result is that the amount of carb required during physical activity is negatively associated with pre-exercise glucose values.

 Some points should be clarified:

 In the introduction

·      it should be mentioned that other approaches have also been explored to avoid hypoglycemia during or after physical activity in T1D patients, as minidose glucagon (DOI : 10.2337/dc18-0051) or sprint at the end of a moderate intensity session (DOI : 10.1007/s00125-014-3218-8).

 In the method section: 

·      the paragraph detailing the maximum cardiopulmonary exercise testing is too long in my opinion, and not in the scope of this article. Please, could you shorten this paragraph.

·      You state that “A stable basal insulin therapy was defined as a pre-breakfast self-measured blood glucose concentration of 4 to 7 mmol/L (72 - 126 mg/dL) over six consecutive days.” with the same basal insulin dose? Please clarify. In addition, was this goal maintained after three days of 25% basal reduction ?

·      “After a minimum wash-out period of a of 4 weeks, participants performed the second exercise period with the remaining basal insulin dosing 186 scheme.” Could you edit this sentence.

·      A figure resuming the study visits and detailing the process of physical activity sessions would be of great help to understand the method.

·      It is unclear how many carb intakes were allowed during physical activity ? only one ? if not, what was the allowed frequency for carb intake ?

 In the results section:

·      In the method, you state that patients received carbo-hydrates prior to exercise session if blood glucose was below 1.26g/L. However, this information is not available in the results. Could you provide the amount of carbohydrate used prior to physical activity in both periods (basal 100% and basal 75%).

·      It is difficult to understand what is presented in the “3.2 diabetes specific results” : is it CGM results for the all period (including resting periods between physical activity sessions) or CGM during the exercise periods exclusively or plasma glucose during physical activity ? Please, clarify.

·      As patients worn CGM around the exercise periods, it would be interesting to show the Time in range and time in hypo during the nights following exercise bouts, for both periods (basal 100% and basal 75%). Please, provide these data.

 In the discussion section:

·      You state that “From a clinical point of view, our results can be used as a first recommendation on how individuals with type 1 diabetes can maintain euglycemia based on orally administered carbohydrates during ~1 hour of moderate-intensity exercise with little bolus insulin on board.”. This sentence should be moderated as your study might not represent a recommendation.

Author Response

This manuscript presents the secondary objectives of a published study that primarily explored the effect of basal insulin reduction on glucose control during physical activity in MDI T1D patients. The secondary objectives presented here was to investigate the amount of orally administered carbohydrates needed to maintain euglycemia during moderate-intensity physical exercise, taking pre-exercise blood glucose levels and the basal insulin dose into account. The main result is that the amount of carb required during physical activity is negatively associated with pre-exercise glucose values.

We want to thank the reviewer for detailing the main outcomes of our study and for supporting to improve the quality of our manuscript.

--------------------------------------------------------------------------------------------------------------------------------------

Some points should be clarified:

In the introduction it should be mentioned that other approaches have also been explored to avoid hypoglycemia during or after physical activity in T1D patients, as minidose glucagon (DOI : 10.2337/dc18-0051) or sprint at the end of a moderate intensity session (DOI : 10.1007/s00125-014-3218-8).

Thank you very much for the comment. We included both suggested manuscripts within the introduction section as following (page 2, line 54):

“A less commonly applied, but highly effective approach to reduce the risk of exercise-induced hypoglycemia faces subcutaneous mini-dose injections of glucagon [9]; in this study Rickels and colleagues showed that 150 mg of glucagon may be more effective than a 50% basal insulin reduction for preventing exercise-induced hypoglycemia and may result in less post-intervention hyperglycemia than a standardized ingestion of 40 g oral glucose tablets. The only known exogenous approach to avoid post-exercise hypoglycemia is by means of a 10-sec maximal sprint after a moderate-intensity exercise session [10]. Via this short bout of high-intensity exercise, counterregulatory hormones adrenaline and noradrenaline increase hepatic glucose production, hence leading to a stabilization in blood glucose concentration.”

--------------------------------------------------------------------------------------------------------------------------------------

In the method section:

the paragraph detailing the maximum cardiopulmonary exercise testing is too long in my opinion, and not in the scope of this article. Please, could you shorten this paragraph.

Thank you for the comment. We shortened the section accordingly.

--------------------------------------------------------------------------------------------------------------------------------------

You state that “A stable basal insulin therapy was defined as a pre-breakfast self-measured blood glucose concentration of 4 to 7 mmol/L (72 - 126 mg/dL) over six consecutive days.” with the same basal insulin dose? Please clarify. In addition, was this goal maintained after three days of 25% basal reduction?

Thank you very much for this comment. The stable basal insulin therapy must have been achieved with the same basal insulin dose. This is now amended as following:

“Participants included in this study were on a stable insulin Degludec therapy (Novo Nordisk A/S, DEN) for least 3 months prior to their study inclusion. A stable basal insulin therapy was defined as a pre-breakfast self-measured blood glucose concentration of 4 to 7 mmol/L (72 - 126 mg/dL) over six consecutive days with the same basal insulin dose.”

This goal was only achieved/tested for the regular (100%) dose, since we expected higher SMBG concentrations during the 25% reduction phase.

--------------------------------------------------------------------------------------------------------------------------------------

“After a minimum wash-out period of a of 4 weeks, participants performed the second exercise period with the remaining basal insulin dosing 186 scheme.” Could you edit this sentence.

Thank you for the comment. We amended this sentence as following:

“After a minimum wash-out period of 4 weeks, participants performed the second exercise period with the remaining basal insulin dosing scheme.”

--------------------------------------------------------------------------------------------------------------------------------------

A figure resuming the study visits and detailing the process of physical activity sessions would be of great help to understand the method.

Thank you very much for this comment. We incorporated a figure as following:

Figure     1. Study flow     chart. IC = informed consent procedure, iCGM = intermittently-viewed     continuous glucose monitoring system, CPX = peak cardio-pulmonary exercise     test, IDeg = insulin Degludec, CHO = carbohydrates, BG = blood glucose.

--------------------------------------------------------------------------------------------------------------------------------------

It is unclear how many carb intakes were allowed during physical activity? only one? if not, what was the allowed frequency for carb intake?

Thank you very much for this comment. We included this information as following:

“During the exercise sessions carbohydrates were administered orally (15 – 30 g) if the participant dropped below a pre-defined threshold of 7.0 mmol/L (126 mg/dL). If the blood glucose concentration did not increase above 7.0 mmol/L (126 mg/dL) within 10 min after this treatment, this procedure was repeated (every 10 min) as often as required.”

--------------------------------------------------------------------------------------------------------------------------------------

In the results section:

In the method, you state that patients received carbohydrates prior to exercise session if blood glucose was below 1.26g/L. However, this information is not available in the results. Could you provide the amount of carbohydrate used prior to physical activity in both periods (basal 100% and basal 75%).

 Thank you very much for the comment. Immediately before the start of the exercise sessions, no additional carbohydrates were given since in all visits the blood glucose concentration was above 126 mg/dL (based on the integrated FreeStyle Libre blood glucometer, that slightly overestimated the real BG values obtained by the EKF lab diagnostic (~ 17 mg/dL) as recently shown by our research group at the ATTD 2019 conference (please see: DECREASED ACCURACY OF THE FREESTYLE®LIBRE GLUCOMETER DURING ENDURANCE EXERCISE IN PEOPLE WITH TYPE 1 DIABETES: HYPOGLYCAEMIA IS THE WEAK SPOT). However, the amount of the last carbohydrates given prior to the start of the exercise testing (> 2 hrs) was comparable between both trial arms. This information is now included as following:

“The amount of carbohydrates consumed at the last pre-exercise meal was similar between both trial arms (100% insulin Degludec dose 62 ± 6 g versus 75% insulin Degludec dose 64 ± 15 g, p = 0.83).”

--------------------------------------------------------------------------------------------------------------------------------------

It is difficult to understand what is presented in the “3.2 diabetes specific results”: is it CGM results for the all period (including resting periods between physical activity sessions) or CGM during the exercise periods exclusively or plasma glucose during physical activity? Please, clarify.

Thank you for the comments. We did now clarify these findings as following:

“During the course of the intervention, total daily dose of insulin was 0.46 ± 0.15 IU/kg bodyweight while total daily dose of basal insulin was 0.25 ± 0.06 IU/kg bodyweight for the 100% insulin Degludec arm and 0.19 ± 0.05 for the 75% insulin Degludec arm. Time spent in specified glycemic ranges assessed via capillary blood glucose concentration (Biosen S-line, EKF Diagnostics, Barleben, GER) during continuous exercise sessions based on the basal insulin dose (75% or 100%) is displayed in Figure 2.”

“Figure 2. Time spent in glycemic ranges assessed via capillary blood glucose concentration (Biosen S-line, EKF Diagnostics, Barleben, GER) during the continuous exercise sessions for all participants. 75% IDeg: 25% reduced dose of insulin Degludec. 100% IDeg: regular dose of insulin Degludec.”

--------------------------------------------------------------------------------------------------------------------------------------

As patients worn CGM around the exercise periods, it would be interesting to show the Time in range and time in hypo during the nights following exercise bouts, for both periods (basal 100% and basal 75%). Please, provide these data.

Thank you very much for the comment. Unfortunatly, since the nocturnal time spent in glycaemic ranges was not defined as an outcome for this secondary analysis and since these specific data are already published (Moser O, Eckstein ML, Mueller A, et al. Reduction in insulin degludec dosing for multiple exercise sessions improves time spent in euglycaemia in people with type 1 diabetes: A randomized crossover trial. Diabetes Obes Metab. 2018;1–8. https://doi.org/10.1111/dom.13534) we can’t duplicate this information. We understand the importance of these data; however, we hope that the reviewer can understand our decision on not showing this information again.

In the discussion section:

You state that “From a clinical point of view, our results can be used as a first recommendation on how individuals with type 1 diabetes can maintain euglycemia based on orally administered carbohydrates during ~1 hour of moderate-intensity exercise with little bolus insulin on board.”. This sentence should be moderated as your study might not represent a recommendation.

Thank you very for the comment. We moderated this sentence as following:

“From a clinical point of view, our results may serve as a first hint on how individuals with type 1 diabetes can maintain euglycemia based on orally administered carbohydrates during ~1 hour of moderate-intensity exercise with little bolus insulin on board.”

--------------------------------------------------------------------------------------------------------------------------------------

Reviewer 2 Report

This randomized, cross-over study was designed to examine how much oral carbohydrate is needed to prevent hypoglycemia during moderate exercise in adults with type 1 diabetes, as well as whether successive days of activity impacted carbohydrate needs (based on two different basal insulin doses). The main limitations of the study are related to the small subject number and other variables that may not have been fully controlled for (such as pre-exercise bolus insulin doses and carb intake).

Specific issues/questions:

Lines 129-132 (also title of Table 1): "Maximum cardio-pulmonary exercise testing" is typically not reached on a cycle ergometer; usually it is only a "peak" test. If one of the test termination criteria was "the inability to maintain 132 a pedaling cadence of more than 50 rpm for 5 seconds" and other "max" criteria were not met, then it was most assuredly a peak test, not a max one.

Lines 160-162: "Participants were informed to inject the last bolus insulin and consume the last pre-exercise carbohydrate-rich meal at least 2 hours before exercise, to reduce the impact of glycemic variability during the course of the exercise sessions." Two hours is likely not enough time for all bolus insulin to have dissipated. It really depends on the dose (larger doses are absorbed more slowly). Was there any attempt to control for the amount of carbs eaten at these pre-exercise times and the amount of bolus insulin taken then?

Lines 165-166: "the midpoint between the first (LTP1) and the second lactate turn point (LTP2) (63 ± 7% VO2max)" Why exactly was this method used to determine exercise intensity? Lactate levels in the blood during physical activity can vary with other metabolic conditions, such as low carbohydrate intake, exercise training, and more. Why not use a percentage of VO2 Reserve or Heart Rate Reserve to more precisely replicate exercise training intensity for each session?

Lines 171-173: "15 – 30 g of oral carbohydrates were given either via glucose gel or fruit juice, which was repeated after 10 minutes if blood glucose did not increase accordingly following the first anti hypoglycemic treatment." Giving glucose gel instead of juice affects how rapidly hypoglycemia resolves. The glycemic index of glucose is 100, whereas the GI of most juices (containing only fructose) is rather low. Choosing one or the other would have affected how much carbohydrate may have been required to raise blood glucose before the start of the activity.

Line 175: "ketone level was measured (FreeStyle Libre, Abbott, UK)" I am not aware of the FreeStyle Libre being capable of measuring ketone levels. Is this a correct reference?

Lines 185-186: "After a minimum wash-out period of a of 4 weeks, participants..." has a typo in it.

Line 250: "Table 3. Pre-exercise blood glucose quartiles" How were the quartiles determined and what were they exactly? This is not clear here.

Line 250, Table 3: Why are CHO doses given with ranges for the 75% iDeg values but with SD for the 100 iDeg doses?

References 1 and 15 appear to cite the same ADA position statement.

Some punctuation issues, particularly around the use of the word "however" in the middle of sentences.

Author Response

This randomized, cross-over study was designed to examine how much oral carbohydrate is needed to prevent hypoglycemia during moderate exercise in adults with type 1 diabetes, as well as whether successive days of activity impacted carbohydrate needs (based on two different basal insulin doses). The main limitations of the study are related to the small subject number and other variables that may not have been fully controlled for (such as pre-exercise bolus insulin doses and carb intake).

Thank your very much for reviewing our manuscript. We tried to improve the quality of this paper based on the reviewers’ recommendations. We will try in further exercise studies in people with T1D to increase the number of participants. The information regarding pre-exercise bolus insulin and CHO intake is now shown in detail.

--------------------------------------------------------------------------------------------------------------------------------------

Specific issues/questions:

Lines 129-132 (also title of Table 1): "Maximum cardio-pulmonary exercise testing" is typically not reached on a cycle ergometer; usually it is only a "peak" test. If one of the test termination criteria was "the inability to maintain 132 a pedaling cadence of more than 50 rpm for 5 seconds" and other "max" criteria were not met, then it was most assuredly a peak test, not a max one.

Thank you very much for this comment. We agree with the reviewers’ comment and changed the wording accordingly throughout the manuscript.

--------------------------------------------------------------------------------------------------------------------------------------

Lines 160-162: "Participants were informed to inject the last bolus insulin and consume the last pre-exercise carbohydrate-rich meal at least 2 hours before exercise, to reduce the impact of glycemic variability during the course of the exercise sessions." Two hours is likely not enough time for all bolus insulin to have dissipated. It really depends on the dose (larger doses are absorbed more slowly). Was there any attempt to control for the amount of carbs eaten at these pre-exercise times and the amount of bolus insulin taken then?

Thank you very much for the comment. We do agree with the reviewers’ comment that there was still bolus insulin on board, especially for higher absolute doses. We did not attempt to control for bolus IOB, however, participants were told to replicate the timepoint of their last pre-exercise CHO intake and dose of bolus insulin injection for both trial arms – this information was documented in the participants’ diaries. The information is now included as following:

The amount of carbohydrates consumed at the last pre-exercise meal was similar between both trial arms (100% insulin Degludec dose 62 ± 6 g versus 75% insulin Degludec dose 64 ± 15 g, p = 0.83). The dose of the last pre-exercise bolus insulin dose injection (100% insulin Degludec dose 6 (4 - 9) IU versus 75% insulin Degludec dose 6 (4 – 8) IU, p = 0.82) as well as the point of time when the bolus insulin was injected until starting the exercise sessions (100% insulin Degludec dose 4h 36 min ± 53 min versus 75% insulin Degludec dose 4h 06 min ± 29 min , p = 0.22) were similar between both trial arms.

--------------------------------------------------------------------------------------------------------------------------------------

Lines 165-166: "the midpoint between the first (LTP1) and the second lactate turn point (LTP2) (63 ± 7% VO2max)" Why exactly was this method used to determine exercise intensity? Lactate levels in the blood during physical activity can vary with other metabolic conditions, such as low carbohydrate intake, exercise training, and more. Why not use a percentage of VO2 Reserve or Heart Rate Reserve to more precisely replicate exercise training intensity for each session?

Thank you for the comment. Percentages of submaximal markers like LTP2  reflect the individual profile of lactate increases, independently of absolute lactate levels. This means even in case (however not seen in our study) glycogen storage was not fully filled during the peak CPX test, the lactate kinetics stay physiologically normal, hence allowing the detection of the first and second lactate turn points. Additionally, we could recently show in 64 people with T1D that the oxygen economy and the HR are altered during CPX testing. Interestingly, this was associated to the HbA1c level (Moser O, Eckstein ML, McCarthy O, Deere R, Bain SC, Haahr HL, et al. (2018) Heart rate dynamics during cardio-pulmonary exercise testing are associated with glycemic control in individuals with type 1 diabetes. PLoS ONE 13(4): e0194750. https://doi.org/10.1371/journal. pone.0194750; Moser et al. Diabetol Metab Syndr (2017) 9:93 DOI 10.1186/s13098-017-0294-1). Furthermore, even in healthy individuals it was shown that using % of VO2max and VO2R translates to an inhomogeneous metabolic response: “The study demonstrates that prolonged endurance exercise at given percentages of VO2max leads to inhomogeneous La responses and, particularly for high percentages, uncertain maintainability among young males of moderate to high aerobic capacity. This holds true even for subgroups with only little variability in VO2max. The findings are likely to be transferable to exercise at fixed percentages of VO2R. Thus, exercise intensities for endurance training and scientific purposes should preferably not be prescribed as percentages of VO2max or VO2R alone” (doi:10.1016/j.jsams.2008.12.626. Exercise at given percentages of VO2max: Heterogeneous metabolic responses between individuals). This information is also reflected in our results, since we prescribed the exercise intensity as % of the LTP2, that translated to an SD when illustrated as % of VO2max of ± 7%. We hope this is acceptable for the reviewer.

--------------------------------------------------------------------------------------------------------------------------------------

Lines 171-173: "15 – 30 g of oral carbohydrates were given either via glucose gel or fruit juice, which was repeated after 10 minutes if blood glucose did not increase accordingly following the first anti hypoglycemic treatment." Giving glucose gel instead of juice affects how rapidly hypoglycemia resolves. The glycemic index of glucose is 100, whereas the GI of most juices (containing only fructose) is rather low. Choosing one or the other would have affected how much carbohydrate may have been required to raise blood glucose before the start of the activity.

Thank you very much for the comment. In our study, the glucose gel consisted of 67% glucose and 33% fructose and the fruit juices consisted of 72% glucose and 28% fructose. This information is now included as following:

“As a consequence, 15 – 30 g of oral carbohydrates were given either via glucose gel (67% glucose and 33% fructose) or fruit juice (72% glucose and 28% fructose), which was repeated after 10 minutes if blood glucose did not increase accordingly following the first anti-hypoglycemic treatment. The amount of orally administered carbohydrates was documented in the case report form (CRF)”.

Out of this small difference we do not expect detectable differences in the PD profile during exercise testing. Additionally, current literature suggests the opposite, showing in detail that the co-ingestion of glucose and fructose faces higher exogenous CHO oxidation rates than glucose during moderate-intensity exercise (1.7 g/min vs. 1.3 g/min) (Glucose Plus Fructose Ingestion for Post‐Exercise Recovery—Greater than the Sum of Its Parts?  DOI: 10.3390/nu9040344).

--------------------------------------------------------------------------------------------------------------------------------------

Line 175: "ketone level was measured (FreeStyle Libre, Abbott, UK)" I am not aware of the FreeStyle Libre being capable of measuring ketone levels. Is this a correct reference?

Thank you for the comment. Yes, the reference is correct. The Freestyle Libre 1 can measure BG, interstitial glucose and ketone levels.

--------------------------------------------------------------------------------------------------------------------------------------

Lines 185-186: "After a minimum wash-out period of a of 4 weeks, participants..." has a typo in it.

Amended accordingly.

--------------------------------------------------------------------------------------------------------------------------------------

Line 250: "Table 3. Pre-exercise blood glucose quartiles" How were the quartiles determined and what were they exactly? This is not clear here.

Thank you very much. We did clarify as following:

“Pre-exercise blood glucose quartiles based on median, 25th and 75th percentile with concomitant carbohydrate consumption.”

Statistics section:

“Data were tested for distribution via Kolmogorov-Smirnoff test. The effect of the pre-exercise blood glucose concentration on orally administered carbohydrates, when stratified for pre-exercise blood glucose quartiles based on median, 25th and 75th percentile, was analyzed via Kruskal-Wallis ANOVA.”

--------------------------------------------------------------------------------------------------------------------------------------

Line 250, Table 3: Why are CHO doses given with ranges for the 75% iDeg values but with SD for the 100 iDeg doses?

Thank you for the comment. As mentioned in the statistics section, all results shown in the manuscript are following the Kolmogorov-Smirnoff test. In case data were not normally distributed the median and IQR were shown, if data were normally distributed the mean ± SD were shown.

--------------------------------------------------------------------------------------------------------------------------------------

References 1 and 15 appear to cite the same ADA position statement.

Amended accordingly.

--------------------------------------------------------------------------------------------------------------------------------------

Some punctuation issues, particularly around the use of the word "however" in the middle of sentences.

Amended accordingly.

Round  2

Reviewer 1 Report

This revised version address all my comments.

I don't have any additionnal suggestions.